# A Systematic Mapping of Translation-Enabling Technologies for Sign Languages

**Luis Naranjo-Zeledón [1,2,*], Jesús Peral [2,*], Antonio Ferrández [2]**  **and Mario Chacón-Rivas [1]**

1 Instituto Tecnológico de Costa Rica, Inclutec, 30101 Cartago, Costa Rica; machacon@itcr.ac.cr
2 Department of Software and Computing Systems, University of Alicante, San Vicente del Raspeig, 03690 Alicante, Spain; antonio@dlsi.ua.es
* Correspondence: lnaranjo@itcr.ac.cr (L.N.-Z.); jperal@dlsi.ua.es (J.P.);
  Tel.: +506-2550-9499 (L.N.-Z.); +34-96-590-3772 (J.P.)

**Abstract:** Sign languages (SL) are the first language for most deaf people. Consequently, bidirectional communication among deaf and non-deaf people has always been a challenging issue. Sign language usage has increased due to inclusion policies and general public agreement, which must then become evident in information technologies, in the many facets that comprise sign language understanding and its computational treatment. In this study, we conduct a thorough systematic mapping of translation-enabling technologies for sign languages. This mapping has considered the most recommended guidelines for systematic reviews, i.e., those pertaining software engineering, since there is a need to account for interdisciplinary areas of accessibility, human computer interaction, natural language processing, and education, all of them part of ACM (Association for Computing Machinery) computing classification system directly related to software engineering. An ongoing development of a software tool called SYMPLE (SYstematic Mapping and Parallel Loading Engine) facilitated the querying and construction of a base set of candidate studies. A great diversity of topics has been studied over the last 25 years or so, but this systematic mapping allows for comfortable visualization of predominant areas, venues, top authors, and different measures of concentration and dispersion. The systematic review clearly shows a large number of classifications and subclassifications interspersed over time. This is an area of study in which there is much interest, with a basically steady level of scientific publications over the last decade, concentrated mainly in the European continent. The publications by country, nevertheless, usually favor their local sign language.

**Keywords:** systematic mapping; sign language; machine translation; gesture recognition; avatar; animation

## 1. Introduction

This study arises from the need to have a broad outlook into sign languages (SL) and their treatment by computational means, motivated by a lack of research evidence that provides structure to this area, which is transdisciplinary by nature. We spot four areas that have to do most clearly with the problem at hand: accessibility, human-computer interaction, natural language processing, and education. This study is particularly relevant because there are no systematic mappings covering all areas that conform translation-enabling technologies, as far as the authors are aware. The importance of the topic led us to formulate this study to understand what the scholar community has contributed in the areas that integrate the processing of sign languages through computers. Figure 1a,b shows, respectively, the results after using a search engine of scholarly documents (Google Scholar) for mappings and reviews. The indexing services of Scopus and Web of Science were also used to corroborate the searches. The search strings were demanding, in the sense that titles were meant to contain the keywords

"systematic [mapping/review]" and "sign language", hence excluding beforehand those titles that were only vaguely related to the object of study. In fact, for systematic mappings we obtained no results, while systematic reviews returned four results, three of them dealing with health sciences topics (with no computational focus) and one of them just recently published [1] that has to do with specifics of teaching LIBRAS (Brazilian Portuguese Sign Language). The results of this study show that the pedagogical approaches and theories used in the planning and construction of tools for LIBRAS are perfunctory.

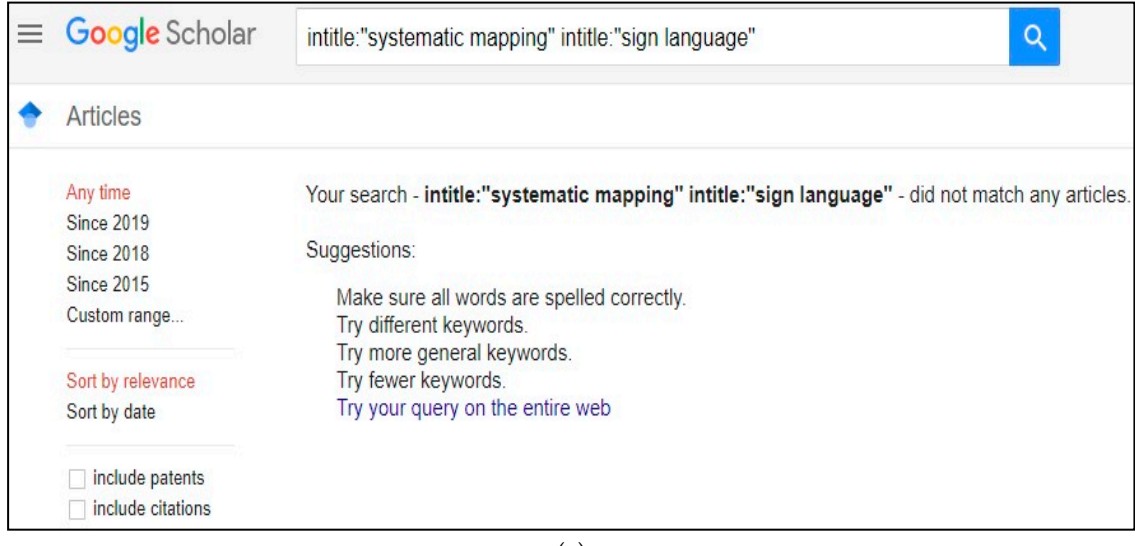

(**a**)

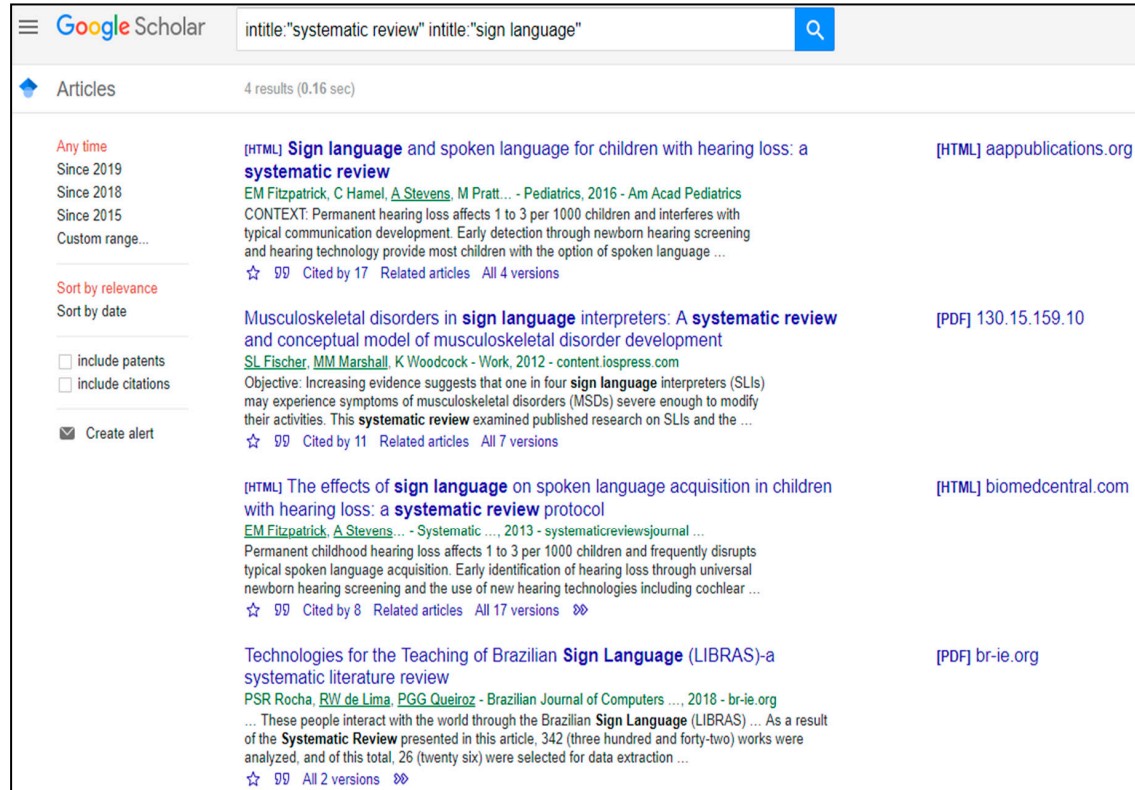

(**b**)

**Figure 1.** (**a**) Search results including keyword "mapping"; (**b**) Search results including keyword "review".

The systematic studies pertaining to medical sciences have to do with the multifactorial elements of musculoskeletal disorder pathologies among SL interpreters [2]. For [3] the aim was to provide a tool for parents, clinicians, researchers, and decision-makers who are looking for evidence in the field of newborn screening, as well as early intervention outcomes, by means of a better understanding of treatments and a timelier introduction of the most effective interventions. On the other hand, a study reported on the outcomes of children with cochlear implants and found that language development was the most frequently reported, followed by speech and speech perception [4].

The lack of systematic studies that integrate the four major areas of interest as well as the appearance of results unrelated to the object of study were both cogent indicators of the need to conduct this research. There is only one result related to the area of interest, specifically addressing education in LIBRAS (Brazilian Portuguese Sign Language), which we will take into account further on in this study.

This systematic study adheres largely to the guidelines suggested by [5,6] and makes the following contributions:

- Provides the scholarly community interested in translation-enabling technologies for sign languages with a broad vision on the subject.
- Quantifies the categories, subcategories, and other relevant criteria that allow sectioning the object of study.
- Displays the results by means of different data visualization techniques.

The remainder of the paper structure is as follows: Section 2 deals with the background and related work, Section 3 explains the research method in detail, Section 4 presents the results of the systematic mapping, Section 5 provides an evaluation of the mapping process, Section 6 discusses the results, and Section 7 concludes the paper.

## 2. Background and Related Work

### 2.1. Sign Languages Overview

Sign languages are powerful means of communication due to their great expressiveness. In many countries such as Costa Rica, Spain, New Zealand, Thailand, and South Africa, they have been declared an official language [7]. In order to achieve a higher level of inclusion, translation systems have been designed between their main users, deaf people, and the rest of the community. Their computational treatment, however, is complex and requires integrating several elements, such as the combination of manual signs with facial gestures, compliance with linguistic precepts, and particularities of the geographical region of the signers. The deaf culture plays a preponderant role in the conceptualization and evaluation of all efforts to generate projects with a broad impact on society.

Sign languages that have undergone a process of maturation have well-defined grammars, parallel corpora, and datasets for the purpose of experimentation. Algorithms are proposed and documented against a baseline to determine their effectiveness. In these efforts, computer scientists, linguists, educators, and members of the deaf community participate in order to deal with the complexities of a phenomenon as broad as that of communication.

The many translation-enabling technologies available for sign languages can be conveniently grouped into four categories [8] that clearly stand out: accessibility, human-computer interaction, natural language processing, and education. The difference between computational linguistics and natural language processing is still a matter of debate, as discussed by [9], and is not the focus of this study. In this case, we will refer generically to this area of study as natural language processing. These areas are by no means exclusive; quite the contrary, they are complementary, and a real system aimed at people with disabilities is expected to adopt these categories in an appropriate manner. A machine translation system must contemplate all these components to produce not only correct translations but also to endow the users and the community around them with elements of success in the solution offered.

### 2.2. Technologies Used in SL Machine Translation

The authors in [10] studied potential technology solutions for e-learning platforms through translation of sign language. They presented a list of potential technology options for the recognition, translation and presentation of SL, as well as potential problems, by analyzing assistive technologies, methods, and techniques. Their analysis shows that some technology solutions are under research and development to be available for digital environments. However, some critical challenges must be solved, and a strong integration of these technologies in e-learning platforms is still missing since there are no immediate solutions to solve synchronous real-time communications between deaf and not deaf people.

Avatars are widely used. [11] developed an agent with a high level of detail that represents gestures in Spanish sign language. In several research departments, an attempt has been made to couple the recognition of gestures with the shapes and movements of hands, arms, and trunk, but it has been reported that the great problem is the construction of an animation.

Some approaches use voice recognition techniques to translate from spoken language into sign language, while other approaches translate written into sign language [12–17]. The authors in [11] note that voice recognition limits its action to specific domains and are not very efficient, at a rate of 8 s per sentence (an impractical solution for real time). Sign languages are natural and evolve over time, which implies a need to update the grammatical basis. Moreover, [18] indicates that grammar and double meaning gestures make translation extremely complicated.

One of the research goals for mapping studies is to determine the necessity to undertake a full systematic review [19]. In this study, this necessity has become quite evident since there is no coherent body of knowledge tying together the many technological components towards high quality solutions.

### 2.3. Applications Currently Available

In addition to the great efforts that have been made within the scholar community, there are already several industry systems available to the public. The role that the industry plays partly feeds on ideas that have emerged in the academy, but additional ideas can also contribute to it and verify or refute arguments that require private investment to be put into practice. The authors wish to emphasize that this section complements the findings of the systematic search coming from the results obtained from a general-purpose search engine, not in academic repositories. In the next subsections the reader will find a description of these systems.

#### 2.3.1. Mobile Applications Already Available

It should be noted that the effort invested by the developers of these proposals is of great importance as they make available to users a range of applications for widespread use through the use of mobile phones.

Hand Talk performs digital and automatic translation into Brazilian Sign Language (LIBRAS) through two main products: a website translator, which makes websites accessible in Libras by inserting a button, and an application that takes text or audio as an input and automatically translates it. Their developers remark that these products are complementary to the work of the LIBRAS interpreters [20].

The purpose of Helloasl is to assist the American Sign Language (ASL) learning process. According to their authors, it enables people to meet and interact in a convenient and enjoyable learning experience beyond the basics [21]. They offer interested people an application and a website both designed for learning purposes.

Visualfy is a product developed by Marc Tamarit and Toni Alcalde, consisting of a network of connected microphones so that deaf people place them in the plugs of their home. Microphones listen to common household sounds and translate them into visual signals so that deaf people can interpret them easily [22].

### 2.3.2. Applications for Web, Windows, and Android

TextoSIGN is a dictionary that converts text into Spanish Sign Language (LSE). The text to be converted is entered into a search box, and after that, a video with an animated 3D avatar is generated. TextoSIGN has about 1500 words, which will increase in future updates. Words can be checked by categories and added to favorites for quick access. There is a lite version, free and limited, and the full and paid version. For now, it works on the Windows platform and is available for Android, but it is planned that it will soon reach the Apple App Store [23].

Signslator is a Flash-enabled website where a text is written and translated into sign language. It is possible to read the words in real time as the user writes them, and it is even possible to change the words in a sentence to gradually learn the sign language. It can translate more than 12,000 words [24].

### 2.3.3. Other Applications

MyVoice is a project of some students from the University of Houston. It is a device in the prototype phase and is responsible for converting sign language to voice. It reads the gestures and symbols of the signing person and translates them into words read to those who do not master that language. The equipment is portable and includes a microphone, speaker, video camera, and screen. The reverse process is also possible with the help of the screen: the user speaks to MyVoice and the equivalent sign language appears on the screen [25].

### 2.3.4. Wearables Incorporating into SL Translation

Hadeel Ayoub, a student at the Goldsmiths Institute at the University of London, developed the SignLanguageGlove, a glove that offers converting gestures into understandable text, displayed on a screen or through a speaker integrated into the glove. It uses five flex sensors, located on each finger, connected to a motherboard that sends the information to a four-digit display or to the speaker [26].

At the Instituto Politécnico Nacional (IPN) in Mexico, a prototype glove was developed to translate sign language into text. After recognizing the signal indicated by the user, the glove sends the information to a mobile phone via Bluetooth so that an application, already published on Google Play, translates it into text. The development is a prototype so it needs improvements, and there are no concrete future marketing plans [27].

### 2.3.5. Real Time SL to Text and Speech

Fundación Vodafone, located in Spain, a mobile phone operator with headquarters in the United Kingdom, presented a proposal called Showleap to facilitate communication between listeners and deaf people. The founders of the project, Teo Atienza and Emilio Guerra, indicate that the software tries to "build the translator based on what deaf people demand". Initially, two bracelets were used that worked well with few signs but failed to increase the database to 20,000 signs. More importantly, deaf people who tried them said that "they didn't want to have to put on anything extra to start a normal conversation". Hence, they decided to change the bracelets for a camera in the user's terminal that detects movements and recognizes images of the person who is signing. Software, which works on the user's mobile, tablet, or laptop, translates the signs in real time and converts them to text and voice. When the hearing person speaks to the deaf person, the application performs the reverse process, converting the words to text for the deaf person to read on their device [28].

### 2.3.6. Systems Incorporating Deep Learning

The aforementioned Showleap uses deep learning techniques, as well as a program that consists of three neural networks: the first processes the video, the second identifies the signs and interprets them, and the third joins the signs and gives meaning to the phrase [29].

Evalk, a Netherlands-based start-up, has developed an artificial intelligence (AI) powered application for deaf people, which promises a low-cost and superior approach to translating sign

language into text and speech in real time. The digital interpreter works by placing a smartphone in front of the user while the application translates gestures and signs into text and speech. The app, called GnoSys, uses neural networks and computer vision for recognition and then translates into speech. Evalk executives state that the translation software in the market is slow or expensive, relying on old technology not suitable to scale to other markets outside country of origin. Their application can be used on multiple devices such as smartphones, tablets, laptops, or PCs. It translates quickly as the person speaks, translates any sign language, and can be plugged into a variety of products, such as video chats, AI assistants, etc. The interpreter for the deaf relies on neural networks. All the translation happens in the cloud. It requires a camera on the device facing the signing person and a connection to the Internet [30].

Students at the Berghs School of Communication in Sweden came up with the idea for the application as a means to enable signed conversation with hearing people. It requires a pair of wrist bands to track the motions of the signing person and send them to the Gesture app. The motions are translated into speech in real time. Using electromyography (a technique for recording the electrical activity produced by the skeletal muscles [31]), it analyzes positions and muscle activity in the hand and forearm. In this way, Google Gesture can identify the signs you are making. A release date for the application has not been announced [32].

Google AI Labs has developed an algorithm capable of tracking the movement of the user's hands after having mapped them with the camera of their mobile. The solution uses machine learning to calculate the 21 key points in three dimensions of a hand within a video frame. To reduce the hardware requirements, they decided to reduce the amount of data the algorithm needs so that the response time is shorter. The position and size of the entire hand is no longer detected but the palm is as it is more distinctive and regular. Then, the fingers are detected.

A total of 30,000 images with different poses and lighting were analyzed. Google AI Labs researchers explain that the novelty of their proposal is that it breaks the current approach, based on powerful desktop environments, with good real-time performance despite working on mobile phones. They claim that they will try to improve accuracy and announce the availability of the source code for other researchers [33].

The Live Transcribe application, from Google, transcribes voice to text in real time for 70 languages, which represents a coverage of more than 80% of the world's population. The application allows to provide automatic subtitles to the conversation that takes place around the user [34].

2.3.7. Systems for Teaching Deaf People to Read

Huawei has joined the European Union of the Deaf and the British Association of the Deaf, in addition to other companies, to create StorySign. It is an application based on artificial intelligence, which reads children's books to convert them into sign language, teaching deaf children to read. The application currently supports ten languages and can run on any Android device version 6.0 or higher. The StorySign application uses Huawei Artificial Intelligence and the mobile camera to detect words. Its operation is simple: once the application is opened, a title is chosen from the StorySign library, and the mobile is held on the pages of the physical copy. The avatar translates the story into sign language according to how the underlined written words are. StorySign supports ten languages (English, French, German, Italian, Spanish, Dutch, Portuguese, Irish, Belgian Flemish, and Swiss German) with one book each. The goal of Huawei is to incorporate many more books in the future [35].

*2.4. Findings and Challenges*

Trends and limitations in sign language translation systems have been evident for the academic community and for the deaf community as well. For both recognition and synthesis purposes, systems normally limited to a particular domain have been developed, such as airports [36–39], train stations [40–43], or hospitals [44–48].

Classical approaches, such as those based on rules, that lead to the creation of dictionaries, with a great knowledge of languages still persist [49]. However, statistical approaches continue to prove extremely effective thanks to the use of parallel corpora between two languages [50]. A disadvantage is that in parallel corpora the phrases of a source or destination language may appear translated into several sentences in their counterpart [51]. Hybrid approaches have been studied that combine rules with statistics and yield very good results [52]. Hybrid systems of rules post-processed by statistics or the inverse approach of rule-guided statistics have been proposed.

With the data collected in this investigation, a significant trend has been detected to prefer rule-based systems, growing intermittently between 2008 and 2018, and statistical-based systems, growing continuously between 2003 and 2013. Systems based in examples or using machine learning techniques still do not represent an important trend in synthesis from written language to sign language, with a slight rebound between the years 2013 to 2017 (from 2013, the methods based on examples have reached a plateau as an option for sign languages). The picture is radically different when it comes to sign recognition towards written language, with a clear trend in the use of machine learning, particularly deep learning with multiple layers [53,54] and to a large extent the use of the statistical approach. Moreover, recent proposals combine deep learning with statistical methods [55,56].

Some studies, such as that of [57], suggest adopting sketch recognition techniques for sign language recognition. In the field of sketch recognition, contributions have been made in grammars and language compilers with good results, which could be experimented in the sign recognition phase [58,59]. This would be an interesting starting point, although there is still much work to do with the treatment of the epenthesis, also known as "movement epenthesis", that occurs between signs while the hands are moving from the posture required by the first sign to that required by the next one [60].

Recognition systems face serious difficulties in capturing the signs in real time, particularly with common use devices such as cell phones, and without requiring the signer to use additional electronic equipment, usually uncomfortable to wear in a day to day setting. In addition, the noise generated by all background images in a real-time environment is an open problem for research purposes.

Synthesis systems, meanwhile, have serious challenges to map text in written language to sign languages, usually reproduced by a signing avatar, mainly because sign languages have a much smaller lexicon.

On the other hand, there are very few contributions in specific research on the management of anaphora [61–63], that is, references to entities in previous discourse, and the treatment of ellipsis or omission of deductible words by context [64–66], which usually lowers the quality of translations.

The disambiguation of words with different meanings has been approached from different perspectives, with partially satisfactory results, mainly with contributions from superficial approaches that have no knowledge of the text but instead apply statistical methods to words that are close to the ambiguous word. Deep approaches assume full knowledge of the word, which presumes a high cost. The superficial approaches, however, have proved more successful [67].

The recognition of named entities is easily resolved only within specific domains. Attempts have been made to broaden its range of action, but the results are much more limited. Even worse, by including recognition methods of named entities, a reduction in the BLEU is frequent [68]. The BLEU metric (Bilingual Evaluation Understudy) is of standard use for machine translation evaluations [69]. Another common metric is WER (Word Error Rate), a measure of the changes needed in the words in a phrase to turn it into another one [70].

Non-standard speech is one of the major limitations in this field of research since rule-based translation, by its very nature, does not include non-standard uses. This causes errors when carrying out the translation process. The construction of parallel corpora with rhetorical language has not been addressed when writing this article.

## 3. Methods

### 3.1. Research Questions

The goal of this systematic mapping study, based on the updated guidelines in [5], is to determine how sign language translation-enabling technologies have been approached since the first known seminal works. Hence, our research questions (RQs) are as follows:

- RQ1: How often the topics of interest have been published?
- RQ2: Which specific topics have been addressed?
- RQ3: Where and when were the studies published?
- RQ4: How were the proposals, implementation, or evaluation processes conducted?
- RQ5: Which proposals have derived in specific products?
- RQ6: What are the research trends and gaps?

This information is then used to synthesize the knowledge base around this subject. Next, we present the devised search protocol.

### 3.2. Search

We have chosen to search in scholarly repositories, based on pre-designed search strings. Another possibility is to start from a known set of articles and from there perform backward snowballing to obtain the articles referenced in this base set. We have opted for the first option since snowballing is most used to extend a review already carried out, and it still has some important features left for further research, like that of identifying a good start set [71].

We use PICO (Population, Intervention, Comparison and Outcomes) as suggested by [6] both to help identify the most relevant keywords and to formulate search strings directly deriving from the research questions.

- Population: In sign languages context, population may refer to specific translation techniques, avatar deployment, application areas, or specific projects. In our context, the population is composed of sign languages, avatars, and translation studies.
- Intervention: In sign languages, intervention refers to methodologies, tools, or technologies. We do not have a specific intervention to be investigated.
- Comparison: In this study, we compare the different proposals, implementations, and evaluations by identifying the strategies used. No empirical comparison is made, but the alternative strategies are identified.
- Outcomes: The number of identified initiatives.

The identified keywords are "sign languages", "avatar", and "translation". These words were not grouped into bigrams since the scope of the study is intentionally left as broad as possible, not even recurring to search modifiers.

This study has been conducted during 2018, after performing an initial web scraping on 2 April. The full years of 2017 and before were considered during the search.

We first launched our query based on an API (Application Programming Interface) provided by [72], which provides access to Google Scholar from Python code, to determine the cardinality of the subject of study. We used an in-house tool still under development, which facilitated to some extent the gathering of the papers by scraping the Web. Nevertheless, not all the studies were publicly available, hence the need to look for directly in the repositories of IEEE Xplore, ACM Digital Library, Scopus, Clarivate, and arXiv in some cases. We did not search the predefined strings in there but looked for specific articles. It is worth mentioning that, as of now, Google Scholar indexes most of the contents of these repositories, publicly available or not, until a few weeks ago in the current year.

To achieve our research purposes, we used the following search string: "sign languages" avatar translation.

*3.3. Data Extraction*

To extract data from the identified primary studies, we developed the template shown in Table 1. Each data extraction field has a data item and a value. The first author performed the extraction, and then the fourth author reviewed it by tracing back the information in the extraction form to the statements in each paper, checking their rightness. Having another author check the extraction is considered a good practice in systematic reviews [73].

**Table 1.** Data extraction form.

| Data Item | Value | *RQ* |
|---|---|---|
| General | - | - |
| Study ID | Integer | - |
| Article Title | Name of the article | - |
| Authors Names | Set of names of the authors | - |
| Year of Publication | Calendar year | RQ3 |
| University/Research Center | Name of the university/research center | RQ2 |
| Venue | Name of publication venue | RQ3 |
| Country | Name of the country (or countries) | RQ3 |
| Characterization | | |
| Sign Language-Project | Name of the sign language or project | RQ3, RQ5 |
| Classification | According to predefined scheme | RQ1, RQ6 |
| Sub-classification | According to predefined scheme | RQ2, RQ6 |
| Abstract | Text | RQ4 |

*3.4. Analysis and Classification*

The information for the extracted items was visually illustrated (see Section 4). The extracted data has been grouped by theme by the second, the third, and the fourth authors during analysis. Then, the papers belonging to each theme were counted.

*3.5. Validity Evaluation*

The following types of validity should be taken into account: descriptive validity, theoretical validity, generalizability, and interpretive validity. The repeatability (or dependability or reproducibility) follows from the previous ones [74]:

- If conclusions cannot be drawn from the data (interpretative validity), it is most likely to draw different conclusions, assuming the research can be repeated
- If there is no generalizability, the study cannot be repeated in different contexts for comparison purposes
- If there are no means to collect correct data, it is likely to get different results when measuring the same attributes.

3.5.1. Descriptive Validity

Descriptive validity is extremely important, no matter whether one is dealing with quantitative or qualitative studies. Nevertheless, the quantitative nature of this study greatly reduces this threat. The primary studies have been kept collected in an online worksheet, in order to perform sorting, clustering, and filtering operations as needed. The worksheet is available as per direct request to the authors for any checking that might be needed. Therefore, we consider that this threat is under control.

3.5.2. Theoretical Validity

The theoretical validity of this study finds its roots in capturing the essence of the object of study. We explicitly explain the possible biases, whenever detected.

Study identification/sampling: By intentionally covering such a large object of study, the use of backward snowballing techniques was impractical. However, the search string used is sufficiently expressive to cover a large number and variety of studies, so we considered that the risk of missing some studies is quite low.

We did not recur to forward snowballing techniques either, but indeed we included newly published studies as we wrote this article, through the Google Alerts system (see Figure 2). Four out of ten studies were gathered a posteriori, and their author lists included four out of ten previously identified top authors. This was done in order to keep the study as up to date as possible, as well as lowering the threat that conducting the extraction of data by one researcher poses on validity.

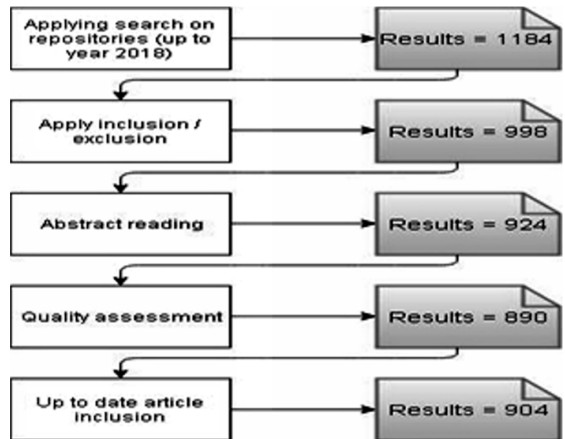

**Figure 2.** Number of included papers during study selection process.

We conducted the study during 2018 and wrote the report during that same year. Studies from 2018 and earlier are included in our analysis. We identified a total of 904 studies, which covered the different areas of interest (see Figure 3).

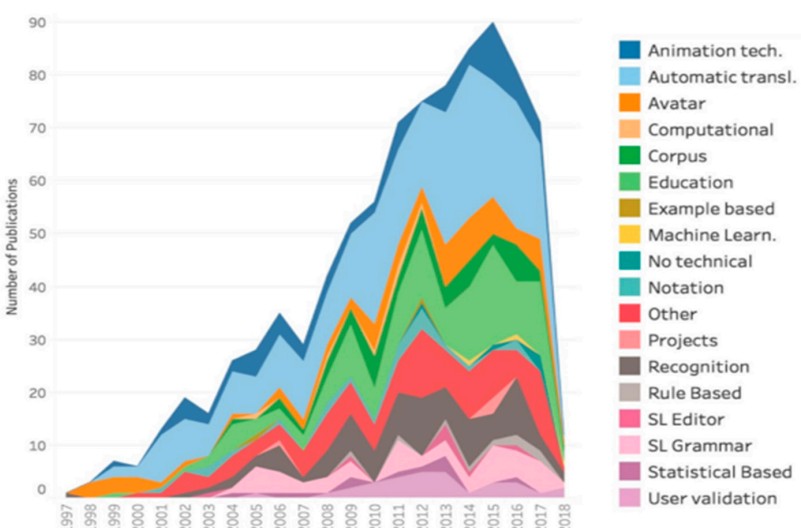

**Figure 3.** Classification-Years area chart.

There is always room for another potential threat since the activities are only those reported by the authors. As a palliative measure, the fourth author checked the extraction.

Data extraction and classification: During this phase researcher bias is also a threat. Authors in [6] indicate that it is useful to have one author extract the data and another one reviewing. To reduce the bias, the fourth author assessed all extractions and suggested new ones. The threat cannot be eliminated, though, since there is human judgement involved.

### 3.5.3. Generalizability

Most identified technologies are addressed recurrently in the literature. There is always, however, the possibility that emerging technologies are poorly represented in a systematic mapping, especially considering that the community often names them in different ways, and it is not until after a few years that there is a consensus on the appropriate nomenclature. In fact, this is not an exclusive feature of our object of study, but it is rather very common in the world of technology, business, and their common areas.

### 3.5.4. Interpretive Validity

We have not detected a bias on the part of the authors in this respect, given that only one of them is co-author of one of the many included studies. Moreover, the practical experience conducting systematic review processes can help in the interpretation of disaggregated and clustered data, hence reinforcing the answers to the research questions and the conclusions of the study.

### 3.5.5. Repeatability

To achieve repeatability, reports must be submitted detailing the methodology followed for systematic mapping. In our case, it is based on the guidelines used by the systematic mapping community. The authors have provided evidence about this process, making the data available to the interested party and have alerted of the possible threats to the validity [75].

## 4. Results

### 4.1. Frequency of Publication (RQ1)

Figure 3 shows the number of mapping studies identified within the years 1996–2018. A few articles that were published most likely prior to 1996 appear undated (six in total). The first dated study was published by [76], the only one in that year. Figure 3 also shows the trends in the different areas that make up the object of study. For the most part, certain constancy or upward trend is noted over the years. Likewise, some gaps are shown, such as that of Rule-based around 2010–2011 or that of Example-based since 2013. These results help answer RQ6, along with what is stated in Section 2.4. "Findings and challenges".

While the interest in these studies was moderately increasing around 2002, a greater increase and diversification can be observed from 2005 on. Besides an increased interest, some important areas like sign language grammars and corpora conformation started emerging around those years.

This evident increase in the number of studies published indicates that this area is considered highly relevant by an ample sector of the research community. In fact, it is not unusual to find systematic studies where the initial set of papers was much bigger than the filtered final results. When that situation happens, this may be indicative of a need to refine the search strings and/or the inclusion and exclusion criteria.

### 4.2. Topics (RQ2)

The topics covered were derived from the ACM Computing Classification System (CCS) [77]. The basic categories were Accessibility, Natural Language Processing, Human-Computer Interaction, and Education. These categories were carefully chosen within the CCS. They were discussed among the authors until reaching a consensus about those categories that better reflected the different areas that make up the object of study, namely, the enabling technologies for sign languages.

Starting from those basic categories, some others arose naturally, for instance "Avatar" and "Translation". Figure 3 shows the magnitude of mapping articles per category in a broad timeline.

It becomes evident that there is an emphasis in "Automatic Translation", "Educational", "Gesture or Sign Recognition", "Avatar", "Corpus", and "SL Grammar". We consider a relevant finding the fact that "Machine learning" is not still as a widespread topic as would be expected when dealing with SL.

The decline noted around 2018 is natural and is due to the fact that it was the current year when most of this research was carried out, particularly the automatic scraping performed in the Google Scholar indexing system.

As expected, there is not only the aforementioned decline, there are others over the years, as well as spikes. The space between the area below and the area above the category analyzed indicates its magnitude for the year or the years to be studied. The interested readers can, in this way, make their own findings. Note that the category "No technical" refers to studies without a computational focus (social impact studies, for instance).

Figure 4a shows a Sankey diagram of the ten most prolific authors in their active period (2007 to 2017). This type of diagram, also known as alluvial, provides the reader with a very comfortable way to visualize flows between two categories of analysis or study variables.

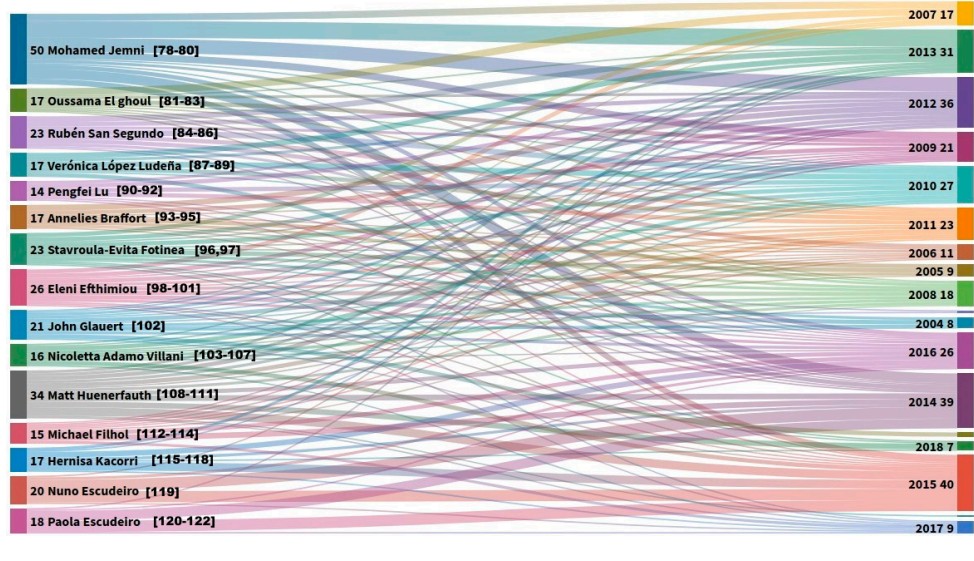

(**a**)

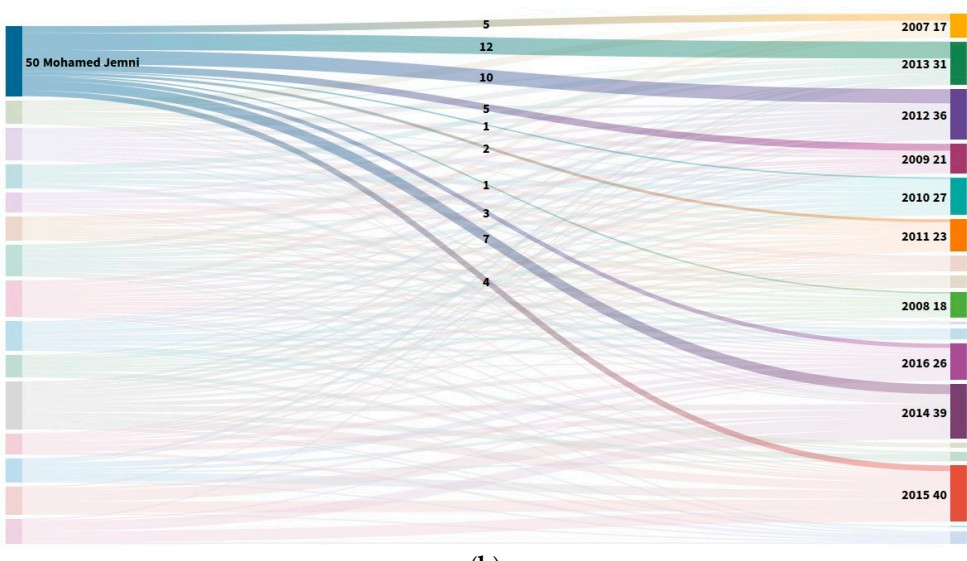

(**b**)

**Figure 4.** *Cont.*

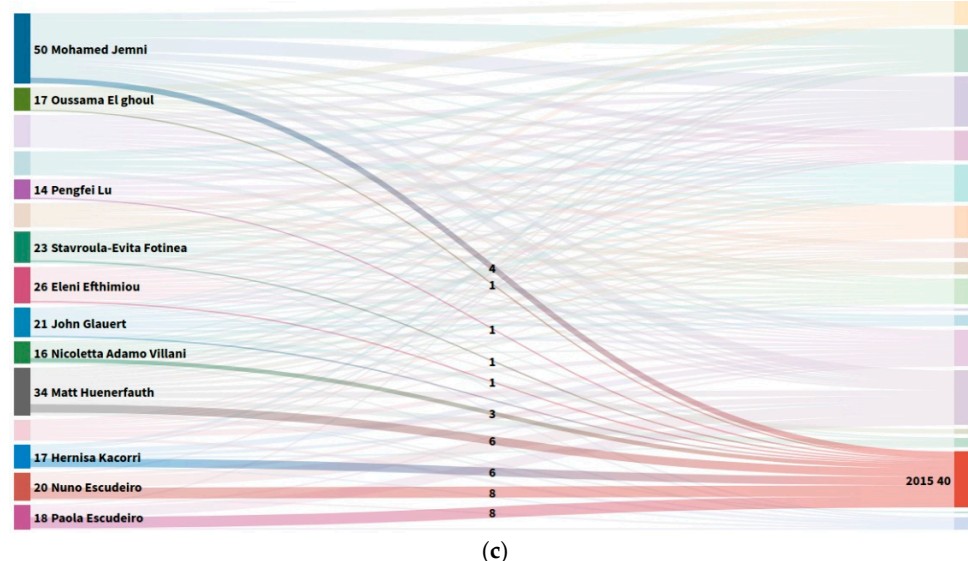

(**c**)

**Figure 4.** (**a**) Sankey diagram among top 10 authors and publication years [78–122]. (**b**) Sankey diagram among top author and publication years. (**c**) Sankey diagram section among top authors and top year.

This particular flow among top authors and publication years seems homogeneously distributed, with the exception of author Mohamed Jemni, who clearly stands out with most publications.

The accumulation of publications for Jemni is also shown in a Sankey diagram in Figure 4b, for a total of 50 works, most of them concentrated in the years 2012 and 2013.

On the other hand, the top publication year is 2015, with 40 articles, followed by 2014 with 39 articles, as depicted in the last Sankey diagram in Figure 4c.

By 2015, the participation of Mohamed Jemni was more balanced compared to the rest of the top authors, with a fairly equitable distribution. Even so, although not the top author in 2014 and 2015, Mohamed Jemni still accounted for 14% of the publications during those two years.

The interactive nature of the alluvial diagrams makes it possible to perform analysis similar to those just presented for each category of analysis by means of a data diagramming tool.

Figure 5 shows a very interesting relation between classifications and subclassifications in a bubble chart that allows measuring the magnitude of this intersection. Some results are predictable, while others are less obvious and striking, such as "Education" and "Notation", with an important quote of works published. On a closer look, this relationship holds for papers dealing with means of interacting for educational purposes, which might come in the expected form of text and avatars or by using standard notations, such as Stokoe [123] or Hamnosys [124], which would be a prerequisite for the communication process.

Another interesting view is portrayed in Figure 6, conceived to relate top 10 authors and classifications. It facilitates to determine that Jemni [125–127], San Segundo [128,129], and López-Ludeña [130–133] are mandatory references when dealing with automatic translation. On the other hand, Braffort [95,134] and Kacorri [135,136], who also integrate the top 10 authors, are more "citable" when dealing with corpus conformation or animation techniques, respectively.

SL grammars, to a lesser degree, have also been addressed, not only by linguists but also by computer scientists.

Table 2 shows a visualization for the top 3 co-occurrence of classifications and their subclassifications (a thorough version of this table is provided in [137]). We assigned each paper a general classification and automatically extracted its possible subclassifications from a list of keywords occurring either in its title or its abstract. This display is indicative, for instance, of the great efforts that have been done in automatic translation, animation techniques, avatars, and recognition. Another

gain from doing this exercise is to corroborate the robustness of the search strings. In our case, "sign language", "translation", and "text" appear with great frequency in the title or abstract.

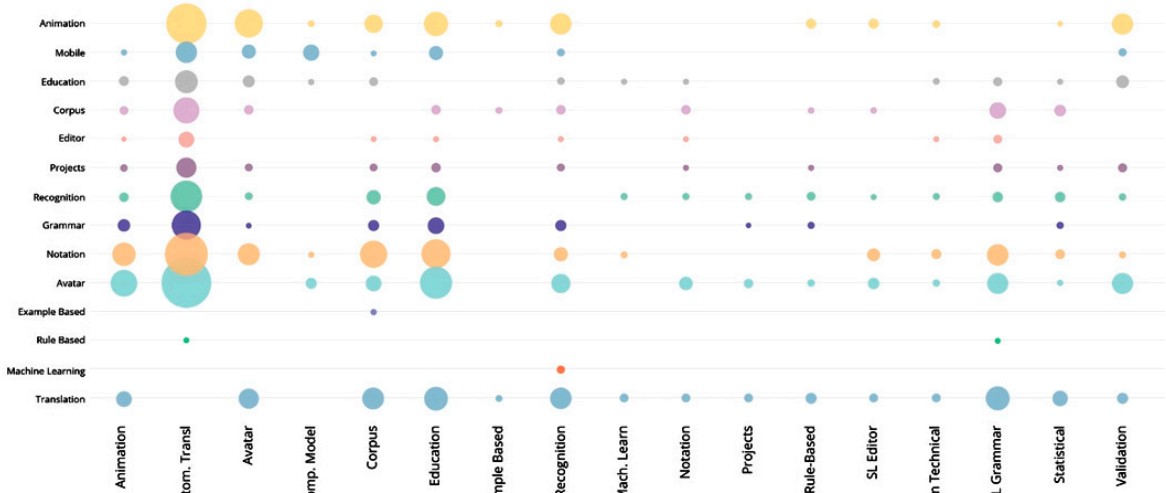

**Figure 5.** Topics covered (classification-subclassification bubble chart).

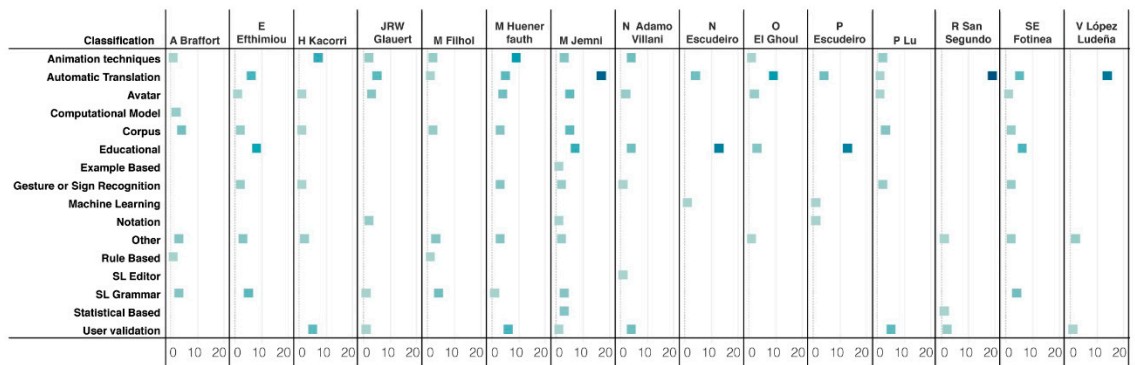

**Figure 6.** Top authors and classification level chart.

Another interesting display is on Figure 7, relating classifications and top authors in a heat map. The reader can see at a glance that the most relevant row in the map is, indeed, "Automatic Translation", and the most important column is "Mohamed Jemni", intersecting in the second cell, with a score of 17. This is only surpassed by author San Segundo in the same category and a score of 19.

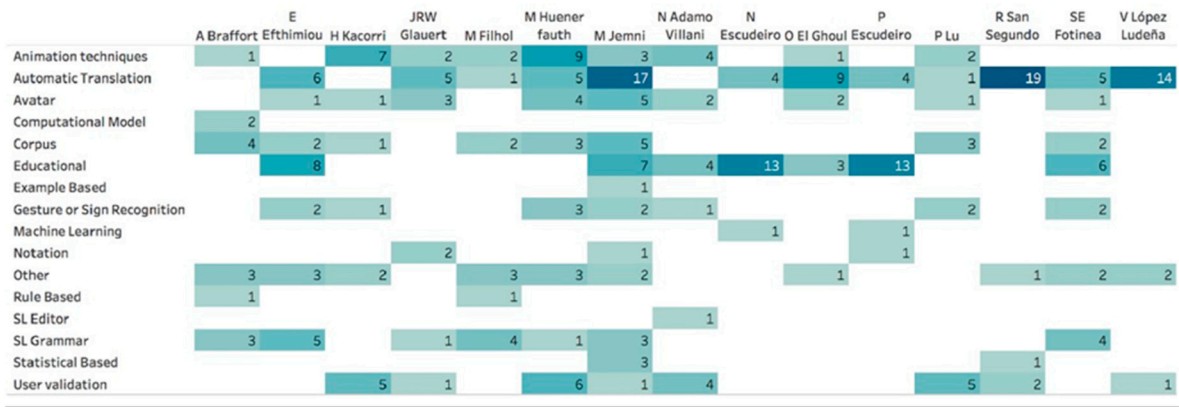

**Figure 7.** Classifications–Top authors heatmap.

**Table 2.** Classification–Subclassification top 3 co-occurrence frequency.

| Classification | Subclassification | Frequency |
|---|---|---|
| Animation Techniques | Avatar | 29 |
| - | Notation | 14 |
| - | Translation | 12 |
| Automatic Translation | Translation | 182 |
| - | Avatar | 104 |
| - | Animation | 68 |
| Avatar | Translation | 18 |
| - | Animation | 32 |
| - | Notation | 32 |
| Computational Model | Avatar | 4 |
| - | Animation | 1 |
| - | Notation | 1 |
| Corpus | Translation | 20 |
| - | Example Based | 1 |
| - | Avatar | 11 |
| Educational | Avatar | 43 |
| - | Translation | 24 |
| - | Animation | 24 |
| Example Based | Translation | 2 |
| - | Animation | 1 |
| - | Corpus | 1 |
| Gesture or Sign Recognition | Translation | 19 |
| - | Machine Learning | 2 |
| - | Avatar | 14 |
| Machine Learning | Translation | 3 |
| - | Notation | 1 |
| - | Recognition | 1 |
| Notation | Translation | 3 |
| - | Avatar | 8 |
| - | Animation | 10 |
| Projects | Translation | 2 |
| - | Avatar | 2 |
| - | Grammar | 1 |
| Rule Based | Translation | 6 |
| - | Avatar | 2 |
| - | Animation | 4 |
| SL Editor | Translation | 2 |
| - | Avatar | 5 |
| - | Animation | 3 |
| SL General-Non technical | Translation | 2 |
| - | Avatar | 2 |
| - | Animation | 2 |
| SL Grammar | Translation | 23 |
| - | Rule Based | 1 |
| - | Avatar | 18 |
| Statistical Based | Translation | 10 |
| - | Avatar | 1 |
| - | Animation | 1 |
| User validation | Translation | 6 |
| - | Avatar | 16 |
| - | Animation | 18 |

## 4.3. Venues of Publication (RQ3)

We have selected a tree map to visualize the venues of publications. Table 3 provides an overview of how the articles map to these venues. International conferences clearly outscore scientific journals. However, these two venues together outscore all others. We have intentionally left the patents in this

visualization, only to demonstrate the low level of claim that exists regarding intellectual property in our object of study, with only six patents granted, all of them in the United States. This situation is shown in Table 4.

**Table 3.** Most frequent venues.

| Venue | Class | Type | Count |
|---|---|---|---|
| Bachelor Thesis | Thesis | Bachelor's Thesis | 5 |
| Book Chapter or Book | Non-refereed | Book Section or Book | 40 |
| Conference Paper | Peer-reviewed | Conference proceedings | 404 |
| Doctoral Thesis | Thesis | Doctoral dissertation | 23 |
| Journal Article | Peer-reviewed | Journal Article | 259 |
| Master–Grade Thesis | Thesis | Master's thesis | 29 |
| Paper–unknown source | Non-refereed conference proceedings | Non-refereed articles | 46 |
| Patent | Patents and invention disclosures | Granted patent | 6 |
| Poster | Peer-reviewed | Conference proceedings | 4 |
| Technical report | Peer-reviewed scientific articles | Conference proceedings | 1 |
| Web Site Project | Unclassified | Unclassified | 2 |
| Workshop Paper | Peer-reviewed | Conference proceedings | 84 |

**Table 4.** Patents granted.

| Authors | Reference | Title | Country | Year |
|---|---|---|---|---|
| WS Jung, HS Kim, JK Jeon, SJ Kim and HW Lee | [138] | Apparatus for bi-directional sign language/speech translation in real time and method | United States | 2018 |
| D Kanevsky, CA Pickover and B Ramabhadran | [139] | Language translation in an environment associated with a virtual application | United States | 2017 |
| D Dharmarajan | [140] | Sign language communication with communication devices | United States | 2017 |
| A Opalka and W Kellard | [141] | Systems and methods for recognition and translation of gestures | United States | 2016 |
| RC Kurzweil | [142] | Use of avatar with event processing | United States | 2015 |
| BR Bokor, AB Smith, DE House, BNII William and PF Haggar | [143] | Translation of gesture responses in a virtual world | United States | 2015 |

The situation depicted is clearly indicative that this kind of studies are regarded as both valid and valuable scientific contributions, since they are widely published in sound scholarly forums.

## 4.4. Approaches (RQ4 and RQ5)

In Figure 8, an area chart is shown comprising research teams and years. There is no clear evidence of sustained predominance over the years; in fact, it is quite variable. The only matter worth highlighting is that authors Matt Huenerfauth from Rochester Institute of Technology and Hernisa Kacorri from University of Maryland have been dominating the scene in the last few years [144–148].

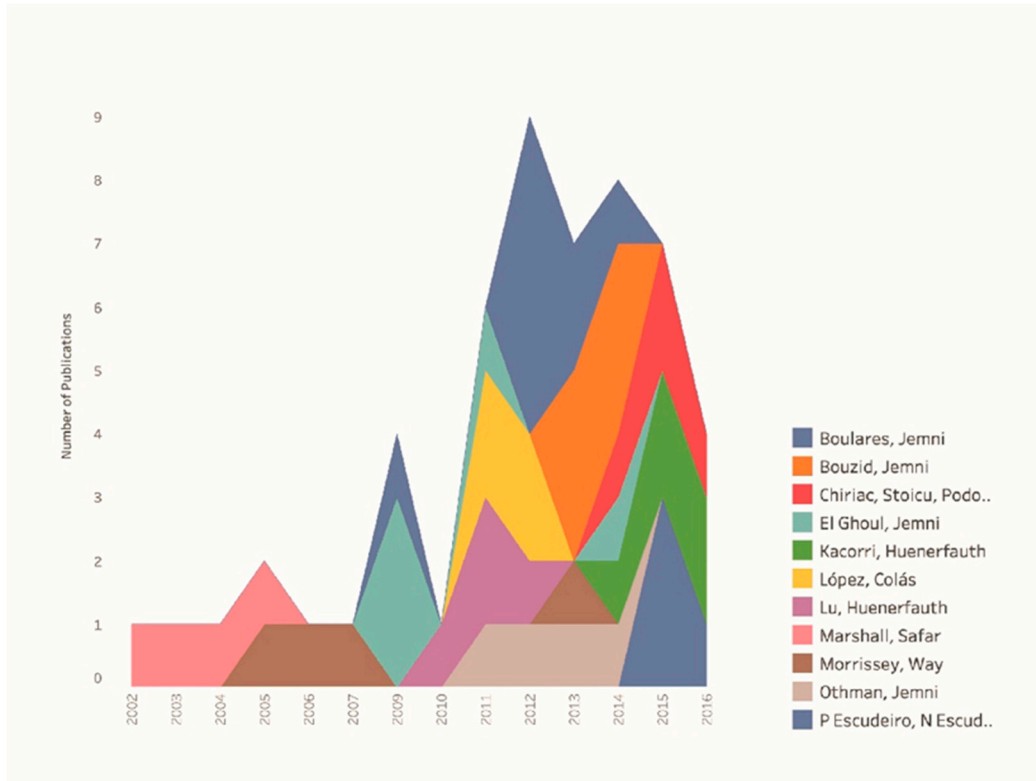

**Figure 8.** Research teams–years area chart.

Paula Escudeiro, Nuno Escudeiro, Marcelo Norberto, and Jorge Lopes, all of them affiliated to Instituto Superior de Engenharia do Porto, Portugal, appear in second place as a joint team of publication in the last few years [149–152].

Figure 9 shows very interesting data, namely, the distribution of studies in specific sign languages by country. The graph shows conclusively that each specific sign language has been addressed mainly in the country to which it belongs (reading horizontally on the graph).

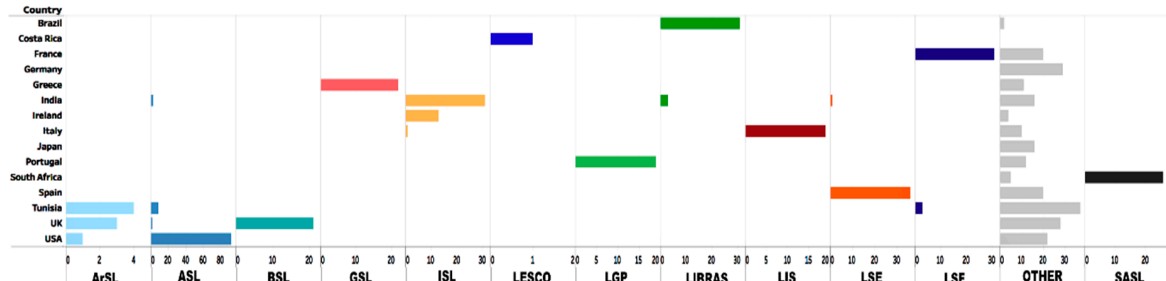

**Figure 9.** Project/SL and country chart.

On the other hand, if a vertical reading is made, it can be seen that the UK, India, and Tunisia have studied not only the sign languages belonging to their country.

In the case of the UK, the British Sign Language (BSL), the American Sign Language (ASL), and the Arabian Sign Language (ArSL) were studied, apart from an important cluster in the "others" group. In India, the research has focused on Indian Sign Language (ISL), the American Sign Language (ASL), the Spanish Sign Language (LSE), and another important cluster in the group "others". In Tunisia, the main focus of research has been American Sign Language (ASL), in second place the Arabian Sign Language (ArSL), in third place the French Sign Language (LSF), and the "others" cluster also appears.

The "others" cluster, by the way, appears practically in all the countries of this study, possibly because researchers have sought to take other languages as a reference to reinforce local studies or because they have better linguistic resources to conduct research.

## 5. Mapping Process Evaluation

In accordance with the good practices recommended by [5], Table 5 shows the relevant actions to be considered in a systematic mapping and those that have been applied in this study, indicated with a check symbol (✔). The symbol "•" represents the actions that were not carried out in this study.

**Table 5.** Activities conducted in this research.

| Phase | Actions | Applied |
|---|---|---|
| Need for mapping | Motivate the need and relevance | ✔ |
| | Define objectives and questions | ✔ |
| | Consult with target audience to define questions | ✔ |
| Study identification | Choosing search strategy | - |
| |     Snowballing | • |
| |     Manual | • |
| |     Conduct database search | ✔ |
| | Develop the search | - |
| |     PICO | ✔ |
| |     Consult librarians or experts | • |
| |     Iteratively try finding more relevant papers | • |
| | Keywords from known papers | ✔ |
| | Use standards, encyclopedias, and thesaurus | • |
| | Evaluate the search | |
| |     Test-set of known papers | • |
| |     Expert evaluates result | ✔ |
| |     Search web-pages of key authors | ✔ |
| |     Test–retest | • |
| | Inclusion and Exclusion | - |
| |     Identify objective criteria for decision | ✔ |
| |     Add additional reviewer, resolve disagreements between them when needed | • |
| |     Decision rules | • |
| Data extraction and classification | Extraction process | - |
| | Identify objective criteria for decision | • |
| | Obscuring information that could bias | • |
| | Add additional reviewer, resolve disagreements between them when needed | • |
| | Test–retest | • |
| | Classification scheme | ✔ |
| | Research type | • |
| | Research method | • |
| | Venue type | ✔ |
| Validity discussion | Validity discussion/limitations provided | ✔ |

The authors considered the "Expert evaluates result" item in Table 5 not only a good practice, as suggested by [5], but a mandatory task to try to eliminate any undetected biases or shortages caused by their own experience in the field. Hence, a total of five work sessions were conducted among the first and fourth authors, and two expert researchers from Aspen University and Universidad de Costa Rica (see Acknowledgments section), both of them dealing with sign language recognition tasks. These forums proved valuable in order to evaluate the whole systematic search process.

Calculating the ratio of the number of actions taken in comparison to the total number of possible actions (12 out of 28) for this mapping, the ratio is 43%, which is significantly above the 33% median

for systematic studies reported by [5]. We want to stress the quality analysis and debugging of data and graphs that we carried out for this study. Eliminating data that does not add value is a process we recommend including in the activities of Table 5.

## 6. Discussion

As for the techniques used, some well-known Natural Language Processing (NLP) tools are used, such as POS-taggers, parsing and to a much lesser extent the resolution of the anaphora or treatment of the ellipsis. Semantic analysis is central to most translation systems. Machine learning has been used much more intensively in recognition than in sign language synthesis. The use of avatars is extremely common for display purposes, as well as voice recognition to produce a translation. The techniques that have been used resort to the use of a predefined corpus to carry out evaluations on the effectiveness of the proposals. Each corpus adapts to the sign language studied, and it can even be adapted to variants by geographical regions. However, it is very difficult to find a corpus endorsed by an organization responsible for regulating sign language. Even in widely studied languages, such as ASL, efforts have been mainly proposed by research centers, notably the CUNY ASL [91,153], or the corpus of the Center for Linguistic Standardization of the Spanish Sign Language (CNLSE corpus) [154].

The most researched languages by the academic community are ASL (American Sign Language), LSE (Spanish Sign Language), ArSL (Arabian Sign Language), LSF (French Sign Language), ISL (Indian Sign Language), LIBRAS (Brazilian Sign Language), SaSL (South African Sign Language), GSL (Greek Sign Language), BSL (British Sign Language), LIS (Italian Sign Language), and LGP (Portuguese Sign Language). The best-known projects are ATLAS (for LSI), AUSLAN (for Australian Sign Language), and WebSign (for ASL). The precision measurements normally used are BLEU with results between 70% and 80% and WER, which is usually between 20% and 30%. Automatic translation systems rely on the use of well-defined grammars at the source and destination or the use of massive data. In this sense, the translation into sign languages does not differ much in concept, but a research and development project may require much more time, due to the limited availability of these resources, including the same formal and normative definition of grammars.

The real-time existing solutions are limited to particular languages and restricted domains, leaving out many communities and areas of relevance. In particular, speech recognition requires very careful treatment and can easily become inefficient. On the other hand, sign languages are dynamic and require updating their grammar bases regularly, which also means a regular update on the software systems that implement them. Ideally, any technique of disambiguation and resolution of ellipsis and anaphora should be considered in every proposal, as well as having corpus labeled to test machine learning techniques. An ideal platform will undoubtedly allow to manage user profiles and adapt to regional variants. The proposals studied show a very clear orientation towards academic projects, which often lack a sustainable financing scheme and are not strongly projected to the community.

## 7. Conclusions

In this systematic mapping study, we found existing literature directly related to technologies meant to facilitate sign languages machine translation. Our evaluation reached the topics investigated, the frequency of publications, the venues of publications, and the specific approaches in use.

The motivation for this study was the lack of a coherent body of knowledge that would provide a comprehensive look into these technologies. In what follows, we answer the research questions of this mapping study.

RQ1, Frequency: The most prolific authors are Mohamed Jemni, Oussama El Ghoul, and Rubén San Segundo, with publications (for most of them) ranging from 2007 to 2018. It was evident that the frequency of publication was often motivated by the ample scope of this field of research.

RQ2, Topics: The topic areas covered were based on the ACM classification and the practical experience of the authors led to subclassifications. The classification for which the highest number of studies has been conducted is "Automatic translation". The classifications "Educational", "Gesture

or Sign Recognition", "Avatar", "Corpus", and "SL Grammar" are also of paramount importance in the field.

RQ3, Venues: More than 80% of the studies have been published in conferences and in reputable journals. We can conclude that these types of studies are considered valuable scientific contributions. The number of studies has increased and kept basically steady since 2005.

RQ4, Approaches: We identified the approaches as well as their application frequency. We followed the suggested evaluation procedure of systematic mapping studies and obtained results above what could be considered the baseline.

RQ5, Specific products: A classification and subclassification co-occurrence frequency display showed that there are seven clearly identified projects (their main assigned classification is precisely a "project").

RQ6, Trends and gaps: An entire section (2.4. Findings and challenges) explains in detail the trends and gaps of the object of study. The trends set a clear course towards new data-centric systems and the hybridization of rule-based with statistic-based. The inclusion of rule-based imposes prior knowledge about the languages of origin and destination, which must be taken into account by the research teams before deciding to use this approach. The gaps have been the same for a long time: mainly the adjustment to well-defined domains, the difficulty to naturally reflect the epenthesis, and an almost total absence of solution of the anaphora and ellipsis.

Since the goal of this mapping study was to provide an overview of the field as broad as possible, we had to make an important effort in gathering a big amount of information. We do not claim that this is always the best course of action since it depends heavily on the particular objective of the research team. In point of fact, comparisons have still to be made by the community on the different strategies of search (repositories, manual search, and snowballing) to determine a reliable way to obtain a sample size.

This study has addressed, aside from academic contributions, industry proposals, some still in the prototype phase. The technologies used so far in the industry for the synthesis and recognition of sign languages show a very clear predilection for incorporating the use of wearables, as well as testing deep learning based prototypes.

In the opinion of the deaf community itself, there is an important dysfunctional aspect which is the fact that the majority of industry proposals focus on wearables, since this prevents conducting a natural conversation, apart from the need of taking care of the device for external use. Sign language synthesis, on the other hand, is reported in the industry as an area that is approached with enthusiasm but is still at an early stage of development, and no benchmarks have been detected that compare their relative advantages and disadvantages. Very desirable characteristics in a synthesis system, such as the treatment of anaphora and ellipsis, have barely been addressed by the academy and are not mentioned in industry efforts.

Finally, the tendency to delimit proposals to specific domains is very clear. Statistical-based and rule-based systems continue to have a leading role, as well as their hybridization, since the requirement for large volumes of data for training continues to represent a gap for many sign languages that do not have large collections for training and testing, the main component of data-centric machine learning approaches.

**Funding:** The authors thank the School of Computing and the Computer Research Center of the Technological Institute of Costa Rica for the financial support, as well as CONICIT (Consejo Nacional para Investigaciones Científicas y Tecnológicas), Costa Rica, under grant 290-2006. The support of our partners from the design department at Inclutec has been crucial to achieve high-quality graphic displays. The feedback of Luis Quesada, Ph.D., from Universidad de Costa Rica and doctoral student Juan Zamora, from Aspen University, regarding adequate form and concept, as well as evaluating the systematic search process, allowed to conceive a definitive version of the paper. This work was partly supported by the Spanish Ministry of Science, Innovation, and Universities through the Project ECLIPSE-UA under Grant RTI2018-094283-B-C32 and the Project INTEGER under Grant RTI2018-094649-B-I00, and partly by the Conselleria de Educación, Investigación, Cultura y Deporte of the Community of Valencia, Spain, within the Project PROMETEO/2018/089.

**Conflicts of Interest:** The authors declare no conflict of interest.

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
