# Peer review of "A Systematic Mapping of Translation-Enabling Technologies for Sign Languages"

_electronics, doi:10.3390/electronics8091047_

Round 1

Reviewer 1 Report

In this work, the authors present a systematic mapping of the research related to Sign Language dating from 1996 to 2018. The authors aim to answer the research questions including the frequency of publications, topics addressed, venue to where the studies were published, implementation processes conducted and what products have been dervied from the past works.

The main strengths of this work are:
1) The paper is very well written and easy to follow
2) The authors have been very systematic and careful in the approach to avoid biases.
3) This systematic mapping can serve the community as an overview of the Sign Language works which can help researchers starting to get broad picture of the field

The main weaknesses of the work are:
1) The authors rely on the Google scholar API and a set of keywords to find the research. This highly depends on what Google returns and might exclude works not indexed or ranked by google. It also has a heavy bias towards academic works.
2) The manuscript gives a big focus on reproducibility but reproducibility highly depends on the results returned by google. For example if the results returned by google change in the future the community cannot reproduce the findings based on the keywords searched.
3) The manuscript is missing a lot of citations and many terms introduced are not cited such as BLEU and WER.
4) The authors mentioned that they check for the validity of results in table 5 by "Expert evalautes result" but have not provided details on what credentials or experience has the expert.

Reviewer 2 Report

The paper presents a systematic mapping of translation enabling technologies for sign languages. The work aims to analyze the predominant areas, authors, and venues in this research field.

I appreciate the effort of the authors to analyze a lot of papers and to summarize their contributions. Nevertheless, I think that as it is now the proposal is quite poor. Thus, authors should make a considerable effort to enrich and improve their proposal to make it publishable. In particular, the paper mainly provides statistics on the considered papers without giving information on current research trends and limitations of existing translation systems.

In what follows I provide some detailed comments on the improvements that in my opinion are necessary:

The research questions in the mapping study should be able to discover (past and current) research trends and research gaps. Thus, authors should introduce fine grained RQs focusing on these two aspects (trends and gaps).

Beyond some work of the specific domain, I think authors should take into consideration and discuss correlations of their proposal with work on gesture/sketch recognition, which are used for communicating between human and machines as well as between people using sign language. In the past decades solutions proposed in this field have led to the development of several recognition systems. For instance, much work has been done in the field of grammars and language compilers, which might also provide authors with some ideas to enrich the study they propose. They can take a look at the following work and connected references:

Costagliola et al., A multi-layer parsing strategy for on-line recognition of hand-drawn diagrams, Proceedings - IEEE Symposium on Visual Languages and Human-Centric Computing, VL/HCC 2006, pp. 103-110.

Costagliola et al., Sketch grammars: A formalism for describing and recognizing diagrammatic sketch languages, Proceedings of the International Conference on Document Analysis and Recognition, ICDAR 2005, pp. 1226-1230.

Table 2 and the table of the appendix make the paper hardly readable, they should be summarized in a figure.

Minor remarks:

Line 20: please remove “We have found”

Lines 286-287: from figure 5 it is not clear why the discussed relationships hold (e.g., Automatic translation and text, Education and Notation)

The quality of figures 6 and 9 is poor.

Line 330: “in recent years” should be “in the last few years”

Round 2

Reviewer 1 Report

The authors have addressed the reviewers concerns.

Author Response

Thank you for the reviewer's comments. We have reviewed the English spell check.

Reviewer 2 Report

I find that the presentation of the paper has improved, and my comments have been sufficiently addressed. The discussion about trends and gaps strengthen the paper.

Minor comments:
- remove widows and orphans
- improve the quality of Figure 2
- Figure 5 is missing
- Line 465: analyzes -> analysis

Author Response

Thank you for the reviewer's comments. We have reviewed the English spell check.

   Furthermore, we have corrected the minor errors suggested by the reviewer.